# Block-operations: Creating an Inductive Bias to Route Data and Reuse Subnetworks

## Abstract

Feed Forward Neural Networks (FNNs) often suffer from poor generalization due to their inability to effectively develop and reuse subnetworks for related tasks. Csordás et al. (2020) suggest that this may be because FNNs are more likely to learn new mappings than to copy and route activation patterns without altering them. To tackle this problem, we propose the concept of block-operations: Learnable functions that group neurons into larger semantic units and operate on these blocks, with routing as a primitive operation. As a first step, we introduce the Multiplexer, a new architectural component that enhances the FNN by adding block-operations to it. We experimentally verified that the Multiplexer exhibits several desirable properties, as compared to the FNN which it replaces: It represents concepts consistently with the same neuron activation patterns throughout the network, suffers less from negative interference, shows an increased propensity for specialization and transfer learning, can more easily reuse learned subnetworks for new tasks, and is particularly effective at learning algorithmic tasks with conditional logics. In several cases, the Multiplexer achieved 100% OOD-generalization on our tasks, where FNNs only learned correlations that failed to generalize. Our results suggest that block-operations are a promising direction for future research. Adapting more complex architectures than the FNN to make use of them could lead to increased compositionality and better generalization.

## 1 Introduction

**Problem**. Typical artificial neural networks (NNs) perform poorly on tasks of systematic generalization (Marcus (1998), Bahdanau et al. (2018), Lake & Baroni (2018)). Numerous people have argued that this failure to generalize is caused by Neural Networks' lack of compositionality: The ability to break complex concepts down into their atomic elements and to reuse and recombine them appropriately when faced with a new task (Pfeiffer et al. (2023), Lake & Baroni (2018), Bahdanau et al. (2018), Barrett et al. (2018), Hupkes et al. (2020), Hill et al. (2019)).

**Compositionality**. Csordás et al. (2020) investigated in how far compositionality emerges automatically in neural networks. They found that neural networks frequently learn to develop specialized subnetworks for different tasks, but have **difficulty learning to reuse subnetworks** for similar tasks. They note that **neural networks are bad at routing data** because routing can only occur through network weights with a special structure that is difficult to learn. Their experiments suggest that neural networks tend to learn new feature mappings instead of representation-preserving mappings even in situations where a uniform representation would clearly be beneficial. They argue that this is an important issue and call for additional research on suitable inductive biases. This paper aims to answer that call, using experiments similar to their own to verify the effectiveness of our module.

**Novelty**. The main novelty we introduce is the concept of block-operations: **We reconceptualize how neurons in network activations are treated**. Where fully-connected layers treat all neurons as independent of each other and where attention-mechanisms linearly interpolate entire tensors, we do a mix of both: **We split activation tensors into uniformly sized blocks**. We then introduce the Multiplexer and SMFR modules (Stack of Multiplexers and Feedforward Neural Networks with gated residual connections), which treat these blocks as semantic units that can be moved, processed and recombined independently. The SMFR learns to apply copy operations and interpolations between blocks, as well as feature mappings within each block. This creates an inductive bias that makes it

very easy for the network to learn to transfer data without losing the generality of fully-connected layers. **The SMFR replaces the basic Feedforward Neural Network (FNN) building block**. It can learn to emulate an FNN or to transfer data, depending on what is most useful.

**Experiments**. We compare the SMFR to the FNN, which it replaces, using synthetic datasets to test directly for the properties we are looking for. Our module showed signs of compositional behavior and attained substantial improvements over the FNN baseline. In Section 5.1, we show that our module suffers less from negative interference (McCloskey & Cohen (1989)). In Section 5.2, we show that our module is better at reusing modules and keeping data representations consistent throughout the network. This improves generalization and enables transfer learning. It achieved **perfect Out-of-Distribution (OOD) generalization on several trials by learning to reuse a subnetwork**. In Section 5.3, we show that our module is particularly effective at learning logical rules and variable assignments, such as those seen in algorithmic tasks. It generalized in a way that suggests that the module learned the underlying atomic operations of the task, unlike the FNN baseline, which only learned statistical correlations.

## 2 RELATED WORK

**Residual Connections**. Residual connections allow a neural network to route data through the network unchanged, and many variants of this have been used to great success in different areas (Szegedy et al. (2017), He et al. (2016b), Jastrzebski et al. (2017), Bachlechner et al. (2021), Wu et al. (2019), He et al. (2020)). Due to the success of residual connections, we adjust them for our own architecture. In particular, we use a Copy Gate using a learnable interpolation weight instead of simple addition, as used by Csordás et al. (2021) in the Neural Data Router (NDR). They reported very strong generalization ability on multiple tasks using this variant of residual connections.

**Attention**. Attention mechanisms have become a ubiquitous tool in deep learning, useful in a variety of fields such as natural language processing, image processing and speech recognition (Bahdanau et al. (2014), Xu et al. (2015), Vaswani et al. (2017), Devlin et al. (2018), Jaderberg et al. (2015), Chorowski et al. (2015)). Note that the term *Attention* is not used consistently in the literature and is often conflated with the more specific *Self-Attention* that is used in Transformers. The Transformer has difficulty learning to route data without modification, just like the FNN, because it includes a multiplication with a Value matrix, which is similar to applying a fully connected layer. The Multiplexer module we introduce in this paper uses a form of Attention, but not Self-Attention, because the format requirements of its input and output are different, and it avoids using a Value Matrix. Despite their seeming similarity, Multiplexers and Self-Attention mechanisms therefore fulfill different, complementary functions in an architecture.

**Other**. Many other architectures for routing exist and have been summarized by surveys (Pfeiffer et al. (2023), Han et al. (2021), McGill & Perona (2017)). For example, **Capsule Networks** can perform routing between a fixed number of capsules based on entity recognition (Sabour et al. (2017)). **Routing Networks** are a family of architectures that learn routing directly through a separate routing module (Rosenbaum et al. (2017), Rosenbaum et al. (2019)). **Recurrent Independent Mechanisms** use multiple largely independent cells that communicate through a bottleneck of attention (Goyal et al. (2019)).

## 3 BLOCK-OPERATIONS

**Motivation**. As noted by Csordás et al. (2020), there is no inductive bias that would cause FNNs to keep activation patterns for the same concept consistent throughout the network. A related issue is that neural networks tend to produce much more dense representations for concepts than biological neural networks. Ahmad & Scheinkman (2019) show that a more sparse representation is more resilient to noise. To solve these issues, we want to construct a new neural network module that avoids these problems. In this way, we attempt to construct a neuro-symbolic system using a purely connectionist approach, as suggested by Greff et al. (2020) as a possible solution to the binding problem. We aim for an inductive bias to satisfy the following design goals with much fewer parameter updates and side effects than an FNN:

## 3.1 Design goals

**Division of Concerns**. The activation patterns of different concepts should use different, non-overlapping subsets of neurons.

**Routing**. It should be possible to easily copy activation patterns throughout the network without modification. Once the network has learned such a routing mechanism, it should work reliably even for OOD data.

**Reuse**. The activation pattern of each concept should be reused consistently throughout the network. Subtasks that rely on the same concepts should represent these concepts with the same activation patterns.

**Resilience to Combinatorial Explosion**. Different subtasks can rely on different subsets of all possible concepts used by the network. Fulfilling our design goal *Division of Concerns* needs to remain possible, even as the number of possible concepts in the network grows. We can achieve this through a mechanism to select only relevant subsets of concepts and drop irrelevant data.

**Role Multiplicity**. It can happen that a subtask requires two concepts of the same type for different purposes (e.g. calculating $f(a, b) = a + b^2$ requires two inputs of type 'number'). In these cases, it must be possible to unambiguously encode which of the items is which, even though they are both represented by the same neuron activation pattern and therefore overlap and interfere with each other.

**Conditional Behavior**. All of the above needs to be optional. The network must be able to resort back to simple and efficient densely-connected layers when appropriate. Importantly, it must be possible to make that decision at inference time, conditional on intermediate results, and not hard coded through network weights.

## 3.2 Approach

**Block-Operations**. To achieve these goals, we propose a novel way to design neural network architectures, by reconceptualizing the way we treat data: **All activation tensors are split into uniformly-sized blocks that aggregate individual neurons into larger semantic units**. Each block of a tensor should come to hold one distinct concept and the position of the block within the tensor should designate its role.

**Replacing the FNN**. We introduce the Multiplexer and other modules in Section 4 and then combine them into the SMFR module. The SMFR is a replacement for the FNN that has a suitable inductive bias to learn to work with blocks. It can seamlessly combine two different ways of processing data: Learning new mappings just like a regular FNN, and conditionally routing blocks of data. Figure 1 illustrates the general idea without going into the details.

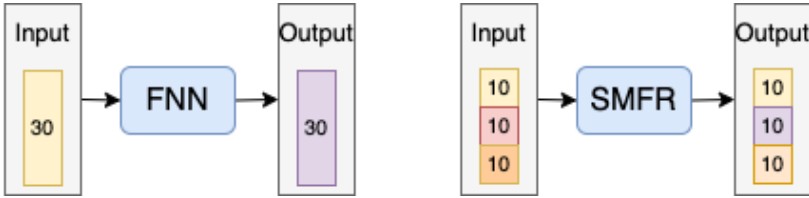

Figure 1: **Left**. An example FNN receives a layer of 30 input neurons and maps it to a layer of 30 output neurons using densely connected layers. **Right.** An equivalent SMFR architecture instead views the input as 3 blocks of 10 neurons each and it outputs another 3 blocks of 10 neurons each. In this example, the first output block is a copy of the first input block. The second output block is generated through an FNN based on all 3 input blocks (the FNN is a submodule inside the SMFR). The third output block is a linear interpolation of the 3 input blocks and an FNN output.

## 4 MODULES

**Overview**. We introduce the **Multiplexer** to route data and the **FNNR** (Feedforward Neural Network with gated residuals) to learn new feature mappings. We combine these two modules into the **MFNNR** (Multiplexer plus Feedforward Neural Network with gated residuals), which can do both. Finally, we stack multiple MFNNRs in sequence to form our final architecture, the **SMFR**.

**Multiplexer**. A Multiplexer (Figure 2) takes $M$ input blocks and produces $N$ output blocks, each of which is a weighted average of all $M$ input blocks. The weights are generated by a Feed Forward network that receives all $M$ blocks as input and outputs an $M * N$ weigth matrix, which is normalized by applying a softmax over the first dimension. The Multiplexer can learn to dynamically transfer blocks based on their content as well as the content of other blocks, can select subsets of blocks or copy them, and can create linear interpolations of different blocks.

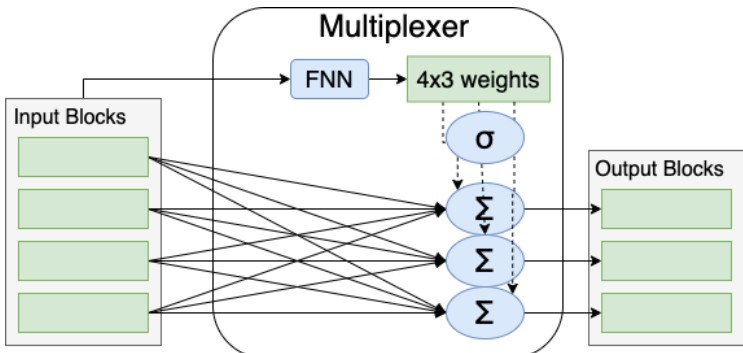

Figure 2: A Multiplexer with $M = 4$ and $N = 3$.

**FNNR**. An FNNR (Figure 3) takes $N$ input blocks and produces $N$ output blocks. It uses an FNN to generate new blocks and then combines them with the input blocks using residual connections. These residual connections use a learned gating weight instead of simple addition, similar to Csordás et al. (2021). An FNNR can learn to either let an input block pass through unchanged, or to replace it with a newly derived block just as an FNN does, or to create a linear interpolation of both. It does this for each of the blocks independently, conditional on each of them as well as on extra input tensors.

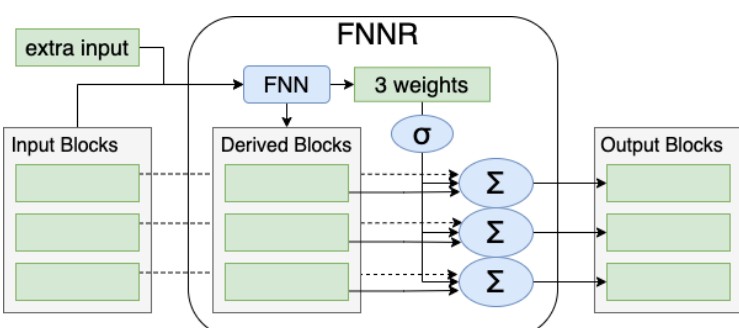

Figure 3: An FNNR with $N = 3$.

**MFNNR**. An MFNNR is composed of a Multiplexer followed by an FNNR module. The FNNR module uses the input blocks of the Multiplexer as its extra input. The MFNNR can learn to rearrange blocks, or to generate new blocks through an FNN, or to interpolate between both. It can learn to do this conditionally on the input and with separate rules for each output block. An MFNNR can emulate an FNN or a data copying mechanism by setting its generated weights to extreme values. It can learn arbitrary mappings like an FNN, but it also has an inductive bias to copy or interpolate blocks of data if doing so leads to a simpler solution.

**SMFR**. Multiple MFNNR modules can be stacked one after the other to form a more powerful architecture. **The SMFR is the architecture we use in our experiments, as an alternative to FNNs.**

**Design Goals**. Coming back to our original design goals: The SMFR achieves *Routing* because both the Multiplexer and the FNNR can pass blocks through without modification. It achieves *Resilience to Combinatorial Explosion* because it can select the subset of blocks that are needed for a task and ignore all others. It achieves *Role Multiplicity* because the Multiplexer can move a block from one position in the input tensor to a different position in the output tensor. It achieves *Conditional Behavior* because both the Multiplexer and the FNNR use an internal FNN to control gates so that all of these features are only used when they are useful. The goals *Division of Concern* and *Reuse* should be achieved as emergent behavior of the network. Our Experiments confirm this assumption.

## 5 EXPERIMENTS

**Summary**. We run experiments on synthetic datasets to test for the presence or absence of useful properties. As the SMFR is intended as a replacement for the FNN, we compare SMFRs with FNNs with a similar numbers of parameters. All experiments are designed to be as simple as possible to test the properties we want to investigate.

**Format**. Each experiment uses a set of digits as inputs, as well as a task indicator to differentiate between the subtasks of the experiment. The digits are one-hot encoded. We use a block size of ten. The task-indicator tensor becomes its own block and is padded to size ten.

**Comparisons and Hyperparameters**. We use grid-search to generate different architectures of FNNs and SMFRs. For FNNs we vary the number and size of intermediate layers. For SMFRs we vary the number of MFNNR modules (depth) and the number of blocks in each layer of neurons between the MFNNR modules (width). A depth of zero means that only a single MFNNR is used to map the input blocks to the output blocks directly. We also vary the size of the FNNs inside the MFNNR modules, but only to a lesser extent. See the Appendix for details.

**Overview**. The experiment in Section 5.1 measures **negative interference** by changing the distribution used for training and testing once a threshold accuracy is reached. It shows that SMFRs are less prone to negative interference than FNNs. The experiment in Section 5.2 measures **generalization through the reuse of subnetworks** by training the same task on two different inputs but with a limited training distribution for one of them. It shows that SMFRs can sometimes learn to reuse subnetworks perfectly even in the absence of training data, a case of transfer learning. The experiment in Section 5.3 measures performance on an algorithmic task that requires **conditional logic and variable assignments.** It shows that SMFRs perform better and develop a more accurate model of the underlying atomic operations.

### 5.1 ADDITION/MULTIPLICATION EXPERIMENTS

**Task**. The addition/multiplication dataset is designed to **test the resilience to negative interference** of a neural network. The task is to either add or multiply two single-digit numbers (modulo 10). This task is similar to the addition/multiplication task in Csordás et al. (2020). We use two training stages. During the preparation stage the multiplication task is trained on limited data and the addition task is trained on all data. In the limited dataset, one number uses only low digits (0 to 4) and the other number uses only high digits (5 to 9). Once the network reaches a threshold of accuracy, we switch to the negative interference stage, in which the rule for training is inverted: The addition task is trained on limited data and the multiplication task is trained on all data. We run the negative interference stage for a fixed number of additional training steps, then measure the OOD accuracy: The accuracy on the data that was only used for training in the preparation stage but not the negative interference stage.

**Variant**. We note that both the addition and the multiplication task are commutative. Our SMFR architecture is good at solving commutative tasks because the softmax used by the Multiplexer is also commutative, giving it a useful inductive bias. This is an additional strength of our architecture over FNNs. We perform an ablation study to measure how much of the SMFR's strength comes from its block-routing abilities and how much from its ability to easily handle commutativity. To do so, we run variants of the experiment in which we use Straight Through Gumbel-Softmax instead of softmax for the Multiplexer, because Gumbel-Softmax makes a discrete pick among candidates and therefore does not help with commutative tasks in the same way softmax does (Jang et al. (2016)).

Table 1: OOD accuracy for different thresholds and architectures

| Threshold | FNN | $SMFR_{Softmax}$ | $SMFR_{Gumbel}$ | $SMFR_{Softmax}/FNN$ |
|---|---|---|---|---|
| 0.7 | $0.032 \pm 0.0034$ | $\mathbf{0.142} \pm 0.0101$ | $0.088 \pm 0.0080$ | **4.44** |
| 0.8 | $0.056 \pm 0.0042$ | $\mathbf{0.184} \pm 0.0112$ | $0.119 \pm 0.0099$ | **3.29** |
| 0.9 | $0.123 \pm 0.0068$ | $\mathbf{0.259} \pm 0.0137$ | $0.208 \pm 0.0113$ | **2.11** |
| 0.95 | $0.202 \pm 0.0105$ | $\mathbf{0.326} \pm 0.0143$ | $0.292 \pm 0.0158$ | **1.61** |
| 1.0 | $0.350 \pm 0.0157$ | $\mathbf{0.434} \pm 0.0179$ | $0.395 \pm 0.0182$ | **1.24** |

**Results**. Table 1 shows the OOD accuracy of different models. Each line compares averages and standard errors of 90 trials of FNN and 63 different architectures of SMFR for each of softmax and Gumbel-Softmax. The Threshold refers to the accuracy at which we switch to the second stage of training. The values show the OOD accuracy 2000 steps after switching to the negative interference stage, except for the last column, which shows the ratio between the values of $SMFR_{Softmax}$ and FNN. We see that $SMFR_{Softmax}$ performs best in all cases and the difference to FNNs is larger for smaller thresholds. In other words, **SMFRs suffer less negative interference than FNNs, especially if the switch between the two training regimens happens earlier**.

**Routing and Commutativity**. $SMFR_{Softmax}$ consistently performs better than $SMFR_{Gumbel}$, and both outperform FNNs. This shows that both the block-routing abilities of SMFRs and their ability to learn commutativity efficiently have positive effects.

**Model Size**. The above analysis is based on an average over all model sizes for both SMFRs and FNNs. An ablation study showed that SMFRs perform better at smaller model sizes and FNNs at larger ones (see appendix for details). The SMFRs were still better than the FNNs at larger model sizes, but the difference was less pronounced. The only cases where FNNs slightly exceeded the performance of SMFRs were when the number of parameters was high and we also waited until convergence ($threshold = 1.0$) before starting negative interference. See the appendix for additional notes on this and on architecture optimization.

## 5.2 DOUBLE-ADDITION EXPERIMENTS

**Task**. The double-addition dataset is designed to **test the ability of a neural network to reuse subnetworks** for similar tasks. The network receives two pairs of numbers and has to return either the sum of the first pair or the sum of the second pair (modulo 10). Like the addition/multiplication task above, this task is similar to the double-addition task in Csordás et al. (2020). We want to measure how well the network learns that both tasks require the same logic. To do so, we use a biased training procedure: The distribution of the input numbers is fully uniform for the first pair of numbers, but it is restricted for the second pair of numbers. This uses the same logic that we also used in the addition/multiplication experiments: One number uses only low digits (0 to 4) and the other number uses only high digits (5 to 9). The training data for the second task is therefore a strict subset of the training data for the first task. During testing, we measure the Out-of-Distribution accuracy on the second pair of numbers. If the network learns both to use the same activation patterns to represent numbers at all layers and to use the exact same process for both tasks, then the OOD accuracy of the second pair of numbers should be perfect.

**Variants**. As before, we run two variants of the SMFRs: One using softmax and one using Straight Through Gumbel-Softmax. Using Gumbel resulted in worse performance overall but otherwise showed the same patterns.

**Results**. FNNs of all model sizes consistently get an OOD accuracy of 0.0, which is actually worse than guessing (0.1). This is because $f(a, b) = a + b$ becomes a bijective function if either $a$ or $b$ are frozen, mapping ten possible inputs to ten possible outputs. The set of possible outputs of the limited training inputs has no overlap with the set of possible outputs of the OOD inputs, which results in an OOD accuracy of 0.0. In contrast, SMFRs reach an average of 0.205 OOD accuracy across all architectures we tested. The surprising part here was that **some trials achieved 100% OOD accuracy**. This is an important finding. Csordás et al. (2020) showed that contemporary neural networks are very bad at learning to reuse logic and our model was sometimes able to do this perfectly.

**Architecture**. We investigated the effect of the architecture on performance. We found that **model size correlated with performance, but bigger was not always better** (see the appendix for details). Across all of our experiments with SMFRs, 10.4% achieved 100% OOD accuracy (25 of 240 trials). Best results were achieved when using softmax instead of Gumbel, at a depth of exactly 1 and a width of 8 or higher: For these architectures **75% of trials achieved 100% OOD accuracy** (9 of 12). These findings suggest that architecture optimization will be important when using SMFRs in practice, which is a limitation that we hope to address in future work. We hypothesize that the depth has an optimum at 1 because at higher depths the network loses track of the routing it should use and just resorts back to using FNN logic.

**Convergence Behavior**. Most experiments with a perfect OOD accuracy converged to 1.0 almost immediately. It happened more often that OOD accuracy increased than that it dropped. Perhaps most importantly, **the OOD accuracy never dropped after reaching 1.0**. In other words, the SMFR usually came to reuse subnetworks more rather than less as the training progressed and remained stable once converged. This suggests that subnetwork reuse will also occur if SMFRs are used as components inside larger architectures that train for longer.

## 5.3 Algorithmic experiments

**Task**. The ALGO task is designed to emulate patterns of information processing that typically occur in real-life coding tasks. It tests how good the network is at **understanding conditional logic and variable-assignment operations**. The task uses five variables as the input and expects the same five variables as the output. The task proceeds in several iterations, modifying one variable per iteration. On each iteration, a formula of the following form is applied:

"Variables $A, B, C, D$ should remain unaltered. Variable $E$ should be assigned the value of variable $A$ if $C > D$ and the value of variable $B$ otherwise."

We use five different permutations of the variables $A, B, C, D$ and $E$, and the task indicator tells the network which of these five rules it should apply on a given iteration. We want to find out if the network learns the actual underlying logic rule, or only a statistical correlate, as FNNs are prone to do. To do so, we use a special way to train and test the network: During training, we always perform exactly two applications of the rule. The network is run for two iterations, and the loss is only applied to the final output. As a result, the accuracy after an odd number of iterations is Out-of-Distribution and tests if the network understands that the data generating process can be decomposed into two applications of the same atomic rule.

**Variant: Incrementation**. In a variant of the task, we also require the overwritten variable to be incremented by one. This additional incrementation step is no extra work for an FNN, but it makes the task harder for an SMFR because it now has to learn a mapping in addition to learning a transfer operation.

**Results**. Figure 4 shows how the average accuracy of the different architectures changes with the number of iterations. Note that the training accuracy is the entry at $iteration = 2$. **The FNNs didn't generalize to odd lengths at all, while the SMFRs only suffered a small loss of performance**. Poor FNN performance was not unexpected, since there is no reason for the FNN to make the results after one iteration human-readable. The SMFR also lost accuracy at odd-numbered iterations, but only to a much smaller extent, which indicates that it learned to decompose the two-step task into the correct atomic logical rule. Besides the average performance, we were also interested in knowing how often an architecture converged to 100% accuracy on the OOD iterations. This is shown by the second graph. SMFRs scored much better here. The majority of SMFR architectures we tried achieved 100% OOD accuracy on odd-numbered iterations. In contrast, no FNN achieved this and most of them even struggled to achieve 100% performance on even-numbered iterations. It appears that **FNNs more easily learned good approximations, but SMFRs were more likely to learn the actual underlying rule that generalizes OOD**.

**Architecture**. The choice of architecture had very strong and consistent effects on performance in this experiment. For the width, all that mattered was making it wide enough that all variables could be represented. For the depth, there was a range of values where all experiments **converged with 100% accuracy on all iterations** regardless of the values of other hyperparameters (The range was 1 to 5 out of 0 to 10). This supports our previous finding that the performance of this architecture

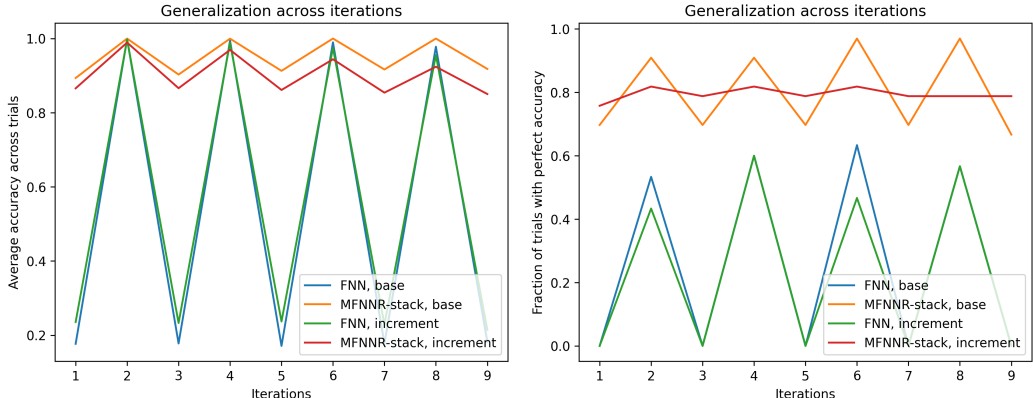

Figure 4: The left figure shows the average accuracy of different architectures and variants at different iterations. The zig-zag pattern is intended behavior: We train on 2 iterations, so odd-numbered iterations are more OOD than even-numbered ones. Note that the five variables become more similar to each other with each iteration if no value is incremented, which is why the performance on higher iterations goes up instead of down for the base variant

can be improved by tuning the number of layers. This is similar to Transformer models, which also achieve greater performance by tuning the number of heads. (Voita et al. (2019)) As in previous experiments, FNNs benefitted more from larger model sizes than SMFRs, and SMFRs reached their best performance at lower model sizes. However, even the largest FNNs we tested performed much worse than the SMFRs on odd iterations (see appendix for details).

**Working with Noisy Input**. In the above experiments, the inputs were all formatted in a way that is easy for the SMFR to work with because the data is already formatted into blocks in the appropriate way: Each block represents one variable. This raises the question: What happens if the input is noisy? Can the SMFR manage to reconstruct the correct format? To test this we ran a second set of experiments with a modification: A permutation is applied to the input, and applied again to the state after each iteration. This permutation is random but fixed, and it permutes all neurons across all blocks. We apply the permutation to intermediate states as well as the input to ensure that any number of iterations can be solved with the same atomic rule. This random permutation had no effect on FNNs, which still give the same performance since they do not care about the order of neurons in a layer. SMFRs likewise did not suffer a reduction on their training accuracy (two iterations) but did lose performance on their 1-iteration OOD-accuracy. However, their OOD-accuracy was still better than that of FNNs on average. Notably, in one of the 30 trials we ran the SMFR did achieve 100% OOD-accuracy again, even though the inputs were permuted. This shows that **SMFRs can learn to automatically arrange unstructured data into the block format they need**. They can sometimes even do so perfectly, but not reliably. Investigating how to do this more reliably remains as future work.

## 6 DISCUSSION

**Block-Operations in Larger Architectures**. The success of our module on the synthetic tasks raises the question: Do these useful properties remain once SMFRs are part of a larger system and applied to practical problems? This remains as future work. The focus of this paper is establishing the soundness of block-operations in general, not solving a practical task. SMFRs are not an architecture of their own but a building block that can serve as a drop-in replacement for FNNs in existing architectures. Since densely-connected layers are so fundamental, it is likely that replacing all FNNs inside large established architectures has side effects. After all, the FNN has been in use for decades and many parts of the ML pipeline, such as the choice of Optimizer and Learning Rate, have been conditioned on the assumption that everything uses FNNs. It remains to be tested if replacing FNNs with SMFR wholesale immediately improves performance or runs afoul of harmful side effects that will require more research to get right. One challenge stands out in particular: Transferring data through an

architecture requires an uninterrupted pipeline, so all modules must allow for *routing* of blocks. Generally speaking, any layer in an architecture that is fully connected can't efficiently learn to pass data through without changing it. All such layers therefore need to be modified. The easiest such modification is the use of residual connections throughout the network. However, ideally modules could instead be adjusted to interact with blocks explicitly.

**Commutativity and Argument Selection**. Besides easier routing, our module also exhibits an inductive bias that helps it to learn tasks that rely on commutative functions, or on selecting arguments from a set of options. FNNs cannot learn these kinds of tasks effectively as they are not permutation invariant. Commutativity and argument selection are fundamental concepts in mathematics and programming respectively, so learning them more easily is likely to be helpful for these types of tasks.

**Interpretability**. In the course of our experiments we noticed that SMFRs were often easier to interpret than FNNs. The activations of the softmax and sigmoid neurons that control the routing correlate with the use of different subtasks. This is analogous to how Transformers can be inspected by visually highlighting how much attention the model pays to different words (Tenney et al. (2019)). These findings are preliminary so far, but promising: They suggest that a correlation anaylsis could tell us which inputs are solved by the same subnetworks and which by different ones. See the appendix for details.

## 7 LIMITATIONS

**Hyperparameter and Architecture Optimization**. SMFRs add hyperparameters to the architecture, which raises the question: What are the optimal block size and the optimal width and depth of the SMFR module? Our experiments showed that simply increasing the depth and width does not always improve performance. Instead, architecture optimization will be necessary to get the best performance out of the model. This used to be a problem for FNNs as well, until He et al. (2016a) fixed this issue by introducing Residual Connections. We hope that a similar fix can be discovered for SMFRs as well and leave this as future work.

**Computational Overhead**. SMFRs take more time per training step than FNNs of equal size because they perform a large number of operations on small matrices. We have found empirically that they take more time than FNNs to converge on simpler tasks but less on harder tasks, where their useful inductive bias outweighs the computational overhead.

**Stability**. We have found that the SMFR architecture can sometimes get stuck in local optima because the gating weights and softmax values take on too extreme values, which kills the gradient. We fixed this by adding a regularization loss: Whenever the absolute of the weight used for a softmax or sigmoid exceeds a threshold value, we apply an MSE-loss to that weight, pushing it back to the threshold (see appendix for details). We have not noticed any other stability problems since then, but as with all new technologies it can not be ruled out that other issues remain that will only reveal themselves on larger tasks.

## 8 CONCLUSION

We introduced the idea of block-operations, a reconceptualization of network activation tensors that aggregates neurons into larger semantic units. Based on this idea, we presented the SMFR, a module that replaces the FNN, with an inductive bias to learn routing and modular decomposition in a neural network more easily. Our experiments confirmed that it can learn to route data more effectively, sometimes even achieving 100% OOD accuracy in tasks that require reusing subnetworks. Our module also proved effective at handling tasks that required understanding commutativity and function argument selection. On an algorithmic task it learned a close approximation of the underlying logic, which is helpful for generalization. Our module can be used as a replacement for FNNs and we expect that it will help for any task that benefits strongly from compositionality. We emphasize that the concept of block-operations that underpins our module design had exactly the effects we predicted. This suggests that block-operations in general may be of interest to other researchers in neural architecture design.

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
