# OpenReview forum: "Block-operations: Creating an Inductive Bias to Route Data and Reuse Subnetworks"
_ICLR.cc/2024/Conference — Submitted to ICLR 2024_

### Official Review · Reviewer_NTVM · 2023-11-01

**Soundness:** 2 fair
**Presentation:** 2 fair
**Contribution:** 2 fair
**Rating:** 5
**Confidence:** 3

**Summary:**

This paper addresses the challenge of poor generalization in Feed Forward Neural Networks (FNNs) by introducing the concept of block-operations. The authors propose a new architectural component, the Multiplexer, which enhances FNNs by incorporating block-operations. They conduct experiments to verify the effectiveness of this approach, comparing it to traditional FNNs. The Multiplexer exhibits several desirable properties, including consistent concept representation, reduced negative interference, improved specialization and transfer learning, and enhanced subnetwork reusability. It excels in learning algorithmic tasks with conditional logics and demonstrates 100% Out-of-Distribution (OOD) generalization in several cases. The paper suggests that block-operations hold promise for future research, potentially improving neural network compositionality and generalization.

**Strengths:**

The paper introduces a novel concept of block-operations, providing a fresh approach to enhance neural network performance.

The proposed Multiplexer demonstrates a range of desirable properties, including improved generalization and transfer learning, making it a practical advancement for neural networks.

The paper's findings open avenues for future research, offering the potential for enhanced network compositionality and generalization in complex architectures.

**Weaknesses:**

It appears that I am not well-versed in this particular research field, but after reviewing the paper, I observed that it may have been hastily prepared. The issues I noticed include inappropriate tables and figures, minor typographical errors, and a lack of overall organization.

The authors mention that "there is no inductive bias that would cause FNNs to keep activation patterns for the same concept consistent throughout the network." This motivation should serve as the crux of the paper, yet it lacks a clear and thorough discussion.

Furthermore, the authors state, "In this way, we attempt to construct a neuro-symbolic system using a purely connectionist approach, as suggested by Greff et al. (2020) as a possible solution to the binding problem." This raises questions about whether the authors are merely adopting an existing solution. What, then, is the unique proposition of this paper? What specific contributions does it make to the field?

**Questions:**

- Could you provide more context and a thorough discussion regarding the statement that "there is no inductive bias that would cause FNNs to keep activation patterns for the same concept consistent throughout the network"? How does this serve as the motivation for your research, and why is it important?

- The paper mentions the attempt to construct a neuro-symbolic system using a purely connectionist approach, as suggested by Greff et al. (2020). Can you clarify how your approach differs from or builds upon this suggestion? What novel elements or contributions does your paper bring to the concept of a neuro-symbolic system?

- Could you provide a clearer and more structured organization of the paper to improve its readability and comprehension for readers who may not be experts in the field?

- What specific issues or challenges in the field of Feed Forward Neural Networks (FFNNs) are you addressing with your proposed concept of block-operations and the Multiplexer? How does this concept enhance FFNNs, and what practical applications or benefits can be derived from it?

- Are there any empirical results or experiments that support the claims made in the paper regarding the performance and effectiveness of the Multiplexer compared to traditional FFNNs? Providing evidence would strengthen the paper's arguments.

- How do the concepts of "block-operations" and "Multiplexer" relate to the broader field of neural network architectures, and what potential implications do they have for various applications beyond the scope of the paper?

- Could you address the concerns raised about inappropriate tables and figures, as well as minor typographical errors in the paper? How can these issues be resolved to enhance the overall presentation of your research?

---

> ### Author Response · Authors · 2023-11-16
>
> **On "there is no inductive bias that would cause FNNs to keep activation patterns for the same concept consistent throughout the network."**: We refer to Csordás et al. (2020) for a thorough explanation.
>
> **On "What, then, is the unique proposition of this paper?"**: The concept of block operations is entirely novel. Greff et al. (2020) made a very generic call for research in this direction, while block operations are a quite specific solution.
>
> **On "there is no inductive bias that would cause FNNs to keep activation patterns for the same concept consistent throughout the network"**: If a concept is not represented consistently then the network can not learn reusable subnetworks, because it it not possible to reuse subnetworks that expect different input formats. Imagine if you wrote a computer program where numbers are represented as int, float, double, etc. at different parts of the algorithm even if they are all integers, so that you end up needing to write a separate function for each of them.
>
> **On "What novel elements or contributions does your paper bring to the concept of a neuro-symbolic system?"**: There are a lot of different possible neuro-symbolic systems. Our main novelty are block-operations. They are unique in that they are designed to satisfy all the goals outlined in section 3.1., which has not previously been done.
>
> **On "Could you provide a clearer and more structured organization of the paper"**: We are uncertain what parts are unclear, but can make improvements that are pointed out to us.
>
> **On "What specific issues or challenges..."**: The goals outlined in section 3.1. can be achieved with block-operations. This makes networks able to learn to reuse concepts more easily and improves modularity. We expect gains in performance and generalization on tasks that rely on modularity.
>
> **On "Are there any empirical results or experiments"**: We focused on synthetic tasks so far because our focus was on proving the soundness of the underlying logic (block-operations), not on beating SOTA with a specific architecture. Our experiments confirm the presence of the properties we were looking for. Running practical tests are next steps in our research.
>
> **On "How do the concepts of "block-operations" and "Multiplexer" relate to the broader field of neural network architectures..."**: Existing neural networks of all types can be modified to make use of block-operations, which may improve their performance on tasks that require modularity.
>
> **On "Could you address the concerns raised about inappropriate tables and figures, as well as minor typographical errors in the paper?"**: We will be happy to fix any errors we find, but is unclear to us in what sense the tables and figures are inappropriate. Could you provide an example?

---

### Official Review · Reviewer_pnJf · 2023-11-05

**Soundness:** 1 poor
**Presentation:** 2 fair
**Contribution:** 1 poor
**Rating:** 3
**Confidence:** 3

**Summary:**

This paper proposes a multiplexer block to replace standard FNNs, to provide a inductive bias in neural networks that encourages re-use and modularity. The paper claims to introduce the concept of "block operations" to split hidden states into blocks, which can then be combined together / routed into different blocks. The multiplexer itself is just a neural network that takes M blocks and outputs N blocks.
The overall model is an "SMFR" block which consists of stacked "Multiplexer + residual FNN" blocks.

The experiments are as follows:
- Addition / multiplication of 2 single digit numbers (mod 10)
- Double addition / multiplication (i.e. the model needs to output two sum / products).
- An algorithmic task which applies conditional rules on 5 different variables, and the objective is to track variable values at the end of these rule applications.

From results, we find that the SMFR model does better than stacked FNNs.

**Strengths:**

Interesting research question on studying / encouraging modularity.

**Weaknesses:**

- The paper lacks soundness (and presentation quality though that may not be a cause for rejection). I'm not sure that the block-operations / multiplexers are doing something qualitatively different from an FNN with residual connection. The behaviour of the multiplexer can very well be simulated by a standard FNN, so i'm not exactly sure what the inductive bias is / where the improvements are coming from. At the very least, it would be good to know how well baselines were tuned / how do results change with slightly bigger models.

- The paper also claims to introduce "block operations" to split activations into multiple blocks. This is definitely not new. For instance, even in ordinary self-attention, hidden states are divided into multiple blocks which are separately consumed by distinct attention heads.

- The specific datasets chosen also seem to be extremely toy-ish for compositional generalization / modularity. Would the proposed approach work on e.g. MNIST (still toy-ish but larger scale than the datasets in the paper).

**Questions:**

N/A

---

> ### Author Response · Authors · 2023-11-16
>
> **On residual connections**: Residual connections do not satisfy the criteria outlined in Section 3.1. as our design goals. They only operate on the entire tensor and can not rearrange parts of the tensor independently of each other. This causes overlap and interference between the neurons used to represent different concepts.
>
> **On model sizes**: We did test models of various sizes and compared performance. As pointed out in the paper, the details on this are in the appendix. Testing performance at even larger model sizes would not be very practical as the tasks are deliberately quite simple and do not require large models.
>
> **On the novelty of block operations**: Other architectures have certainly used mathematically similar operations, but not for the same purpose. Please read Section 3.1. to understand the purpose of these operations. Saying that block operations are not new is like saying that self-attention was not a novel concept when it was first introduced because the softmax function had been used by other papers before.
>
> **On the example tasks**: The tasks are deliberately simple in order to test for the presence of certain properties, which we experimentally confirm. Testing MNIST would not be conducive to our goals: MNIST is a task that relies largely on statistical correlations and requires no reuse of concepts in different contexts. SMFRs help specifically for tasks that rely on modularity. We have run tests on MNIST before and got the results we expected: The SMFR learns not to use routing and learns to simulate an FNN instead. It ends up with the same performance as an FNN. SMFRs only achieve better performance on a task where modularity is useful.

---

### Official Review · Reviewer_frtd · 2023-11-07

**Soundness:** 2 fair
**Presentation:** 2 fair
**Contribution:** 2 fair
**Rating:** 3
**Confidence:** 4

**Summary:**

The authors propose two new layers, the Multiplexer and the FNNR. The multiplexer can linearly interpolate between its input blocks, thus acting as a learnable permutation. The FNNR is a learned transformation with a gated recurrence. The goal of the model is to provide a way of internal routing within the state of the models, as a remedy for their lack of reusing the parameters.

**Strengths:**

- Important problem
- The authors propose a novel architecture to facilitate reusing of subnetworks in NNs
- Interesting algorithm design

**Weaknesses:**

- The main weakness is the quality and design of the experiments, which failed to convince me about the validity of the architecture, and the clarity of the described method.
- The arithmetic tasks are all using 1-digit numbers. This limits the total number of possible training examples in the order of 100. This can result in very easy memorization, and it is also unclear if it provides enough "motivation" for the network to learn to route. The simplicity could be an explanation for why the difference is relatively small between FFN and SMFR in Tab 1. The original paper by Csordás et al had 2-digit inputs, which resulted in ~10k possibilities, which is significantly higher.
- The authors sometimes report performance on all hyperparameters averaged together. This unnecessarily presents their results weaker than they are, and it also makes it harder to understand the actual performance gain. I don't see the utility in averaging in bad hyperparameters. For example at the end of P6, they report an average of "0.205 OOD accuracy", and then "...For these architectures, 75% of trials achieved 100% OOD accuracy". So what is the mean accuracy of the architecture with the best hyperparameters then? Could you please report the mean (and if possible the standard deviation) on the best hyperparameter configuration, as usually done in the literature? Similarly, on page 7, it says "there was a range of values where all experiments converged with 100% accuracy on all iterations", yet in Fig 4 I don't see any method that is 100% on all iterations.
- The claim that "SMFRs can learn to automatically arrange unstructured data into the block format they need" needs more support. Having 1 successful run out of 30 is not convincing enough. For example, input blocks could be decoded back with a linear problem after a single step to show if the architecture managed to disentangle the inputs.
- Some details are unclear: For example, if the weights are shared in the Multiplexer, FNNR, and inside the "Derived Blocks" in FNNR. This should be described by equations in the main paper.
- In FNNR block in Fig.3, and in the description below it, it mentions that "the FNNR module uses the input blocks of the Multiplexer as its extra input". What are the standard, non-extra inputs? What is the *output* of the Multiplexer used for?

**Questions:**

- How are the inputs fed to the network? How are the outputs read out?
- In the explanation of why SMFR performs better on Addition/Multiplication, at the bottom of P5, it says: "... because the softmax used by the Multiplexer is also commutative". What does this mean? The output of the softmax depends on the logits, and those on the specific input blocks unless the gating weights are shared between them. This does not seem to be the case, as far as I can tell from Fig 2 and 3, and the related text.
- The architecture reminds me of a transformer with weights that are not shared between the columns, and with the embedded input chunked in multiple initial hidden states. Can you comment on this similarity?

---

> ### Author Response · Authors · 2023-11-16
>
> **On the experiments**: The experiments are deliberately designed to be a minimal test of the attributes we want to test. Our goal is to test if the concept works as intended, not to beat SOTA. You are correct that the arithmetic tasks are very simple and can be solved largely by memorization. We did also run tests on 2 digits and noticed the same patterns, but since these take much longer to run we focused on the 1-digit version afterwards. While 1-digit arithmetic can more easily be solved with memorization than 2-digit arithmetic, this does not actually affect the validity of our results. Routing and module reuse are just as relevant for memorization tasks as for other types of tasks.
>
> **On how to report performance**: You criticize that we present our results as weaker than they are. Isn't that a good thing? Most papers report the best hyperparameters because it makes their results look better, but this also contributes to replication issues. We deliberately took a pessimistic view, under the assumption that in real-life tasks you won't be able to test all possible hyperparameters. We show that performance is good even so, when averaging over many hyperparameters.
>
> **On "SMFRs can learn to automatically arrange unstructured data into the block format they need"**: Note that it is possible for the network to disentangle the meaning of the unstructured data without being exactly in the format we looked for. For example, by multiplying every value with -1. The "1 of 30" is therefore a lower bound on the architectures ability to rearrange the data. Frankly we were positively surprised to see even the 1 out of 30. The additional layer for linear decoding you propose is a good idea and we will look into it in the future.
>
> **Weight sharing**: No weights are shared. Each of the blue blocks marked "FNN" in the image is just that: A single FNN.
>
> **inputs/outputs**: The Multiplexer takes M blocks and produces N blocks. The FNNR takes the concatenation of both the M original blocks and the N blocks produced by the Multiplexer as input. In this way it can condition both on the original input before interpolation and reordering and on the interpolated blocks produced by the multiplexer.
>
> **Questions**
>
> **input/output format**: The SMFR replaces the FNN and consequently has the same format, with one exception: The input and output tensors' number of neurons must be a multiple of the block size. The SMFR internally treats the tensor as a concatenation of multiple blocks. Any additional dimension such as batch size, iterations, self-attention, etc. all depend on the architecture the SMFR is embedded in. This is analogous to an FNN, whose input dimension is fixed size but which can get additional dimensions based on the architecture it is embedded in.
>
> **On "... because the softmax used by the Multiplexer is also commutative"**: We mean the following: If the network learns (for example) to calculate the softmax as f(x,y, y)=0.5*x+0.5*y+0.0*z where x and y are some blocks, then this function will be commutative with respect to x and y. This would allow the network to solve commutation-based tasks even for OOD data, i.e. even for values of x and y that were not seen in that order during training. While this is a good thing, it is not what we want to test. We therefore make it harder for ourselves by using gumbel-softmax, which does not have this property. This lets us be more confident that the performance gain indeed comes from the routing abilities of the SMFR.
>
> **On transformers**: The SMFR does have a lot of similarity with transformers. However, the similarity is mostly just in the math involved. Their roles are quite different. Transformers are integrated into larger architectures alongside fully connected layers. These fully connected layers destroy the ability to route efficiently, and that is what SMFRs set out to achieve. Even though transformers are internally similar to SMFRs, the FNN layers between them lead to a very different inductive bias.

---

> > ### Comment · Reviewer_frtd · 2023-11-17
> >
> > I would like to thank the authors for clarifications on the input format and weight sharing. However, my main concerns remain unaddressed.
> >
> > > Our goal is to test if the concept works as intended, not to beat SOTA … While 1-digit arithmetic can more easily be solved with memorization than 2-digit arithmetic, this does not actually affect the validity of our results.
> >
> > I agree with the authors that beating SOTA is not the goal, and I don’t think it is necessary by any means for papers introducing novel methods like this. However, I am very skeptical to draw any conclusion from problems where the total number of possible input combinations is in order of 100.
> >
> > > You criticize that we present our results as weaker than they are. Isn't that a good thing?
> >
> > Depends on the specific case. In general not overturning is a good thing. However, mixing all tested hyperparameters makes it impossible to judge how good the method is. What do these numbers mean like this? Let’s say we add another 1000 configurations with “invalid” hyperparameters (like a learning rate of 10000 or 0, or a layer size of 1), the accuracy of each task would go ~0. If the authors are concerned about hyperparameter sensitivity, they could have done instead a sensitivity analysis, or just report separately the best configuration and some distribution of hyperparameters around that best point.
> >
> > > On "SMFRs can learn to automatically arrange unstructured data into the block format they need"
> >
> > I agree that having even a single run is better than having 0. However, claiming that “SMFRs can learn to automatically arrange unstructured data into the block format they need" based on this is an exaggeration.
> >
> > >  If the network learns (for example) to calculate the softmax as f(x,y, y)=0.5x+0.5y+0.0*z where x and y are some blocks, then this function will be commutative with respect to x and y.
> >
> > It is clear that the multiplexer can learn to swap inputs, but it is still unclear to me how any property of the softmax is related to this. The network still has to learn to feed the right logits in the softmax to swap the inputs to the order seen during the training. This is not trivial at all, especially if it never had to swap anything during the training.
> >
> > > Transformers are integrated into larger architectures alongside fully connected layers
> >
> > I’m not sure what the authors meant by this. They are typically used with just an embedding and output classification layer, which is mostly equivalent to how the architecture proposed by the authors is used. If you divide the input vectors such that the input of every block is a separate token, and the columns of the “transformer” are not shared, then the two become very similar.

---

> > > ### Author Response · Authors · 2023-11-17
> > >
> > > We would like to thank the reviewer for the feedback.
> > >
> > > **1-digit arithmetic**: We agree that a low task complexity like this would normally be cause for concern, but we believe that it is not important in this case: For experiments 1 and 2 we care about interference and reuse between tasks and the complexity of the task should not influence greatly how well the network is able to do this. We did run a few trials with 2-digits and those did confirm that we get the same results. We did not run enough trials with 2 digits to report them, because those experiments take longer to run. We can run some more trials with 2 digits if necessary, but we will only be able to run a small number of experiments by the deadline.
> > >
> > > **Presenting the results as weaker than they are**: You raise a valid point. We chose all our hyperparameters to lie in a range that seemed intuitively sensible to us, under the assumption that that's the distribution you could expect when using SMFRs as part of larger architecture, we you do not have time to fine-tune them separately. We did in fact perform a sensitivity analysis, but we were unsure how best to present the results and went with a different presentation instead. Note that we did report both the best parameters and the distribution of hyperparameters around that point: For several experiments we note parameters for which 100% performance is achieved. Maybe it would help if we add more raw data to the appendix?
> > >
> > > **"SMFRs can learn to automatically arrange unstructured data into the block format they need"**: They CAN do so, in the literal sense. If it happened once, then it is possible. Additionally we think it likely that some of the remaining ones of the 30 trials were also rearranged correctly, but not in the way we measured (1-in-30 is a lower bound). Although we agree that the phrasing is misleading because it seems to imply that it's a more frequent occurrence. We could add a "sometimes" to that sentence.
> > >
> > > **On the softmax and swapping**: It suffices if the softmax learns a simple rule that assigns equal weights to two of the numerical inputs and zero to the remaining ones. The output will then be a commutation-invariant interpolation of both inputs and the FNN that uses that interpolation as an input can learn both the addition and the multiplication function rather easily based on this tensor. For example, if input A is 2 and input B is 4, both one-hot encoded, then the interpolation would be the tensor [0, 0, 0.5, 0, 0.5, 0, 0, 0, 0, 0]. We have inspected the network and empirically verified that this does happen on some trials.
> > >
> > > **Transformers**: We believe there was a misunderstanding and we now understand what you mean. Instead of e.g. a SMFR with 5 blocks, you would use a Transformer with input length 5. Correct? In principle a Transformer like this would be able to emulate a Multiplexer, but it would be less effective at it. The main difference here is that the Transformer uses self-attention where each input affects each other input, while the Multiplexer uses a fixed-size neural network and a fixed number of blocks. This makes it easier for the Multiplexer to use each block for a distinct role, because the index of each block is clear, without a need for positional encodings. Additionally, since all blocks are fed into a single NN instead of only having pairwise interactions in a self-attention mechanism, it is easier for the Multiplexer to learn a good logic for routing that takes all input blocks into account.

---

> > > > ### Comment · Reviewer_frtd · 2023-11-20
> > > >
> > > > > 1-digit arithmetic...
> > > >
> > > > I still retain my doubt about concluding anything based on experiments with ~100 possible inputs. Please run some experiments with 2 digits.
> > > >
> > > > > Presenting the results as weaker than they are
> > > >
> > > > I doubt that from the current form of reporting it is possible to conclude anything. I would recommend reporting in the main paper the best hyperparameter configuration for all experiments (preferably with at least 3 seeds and reporting both mean and std), and adding any hyperparameter-related sensitivity analysis to the appendix.
> > > >
> > > > > "SMFRs can learn to automatically arrange unstructured data into the block format they need"
> > > >
> > > > Adding "sometimes" sounds good.
> > > >
> > > > > "On the softmax and swapping"
> > > >
> > > > I see now what the authors meant. Could the authors measure how often this happens? E.g. fix an epsilon, let's say 0.1, and measure how often generalization is linked to the scores of the two inputs being indistinguishable. Moreover, it is unclear to me where this bias is coming from, given that the weights are not shared between the different input blocks. It might be merely a coincidence that leads to better generalization.
> > > >
> > > > > "Transformers"
> > > >
> > > > Correct. Adding a discussion like this to the paper would be interesting (although by no means is a priority compared to the improved reporting of the results discussed above).

---

### Official Review · Reviewer_VWus · 2023-11-08

**Soundness:** 2 fair
**Presentation:** 1 poor
**Contribution:** 3 good
**Rating:** 3
**Confidence:** 4

**Summary:**

The paper proposes a new neural structure called “block operation”, that groups activations into atomic sections, and performs actions over these sections, and studies its empirical impact on the network’s performance on algorithmic tasks.

**Update**:
Dear authors, thanks for the comments. Some responses to them:

**Writing/organization**: of course there could be preferences for different people on that, but personally I think it goes beyond writing style and damages the paper's clarity. It makes explanations kind of shorter, less detailed, and more disconnected. E.g. in slides when things are organized in a way more similar to that, the speaker does the job of making the presentation coherent, and so in the context of a paper, I personally believe this is less suitable / damages it. Indeed, you agree that you would like to add more details but couldn't due to page limit. I think more efforts on making the writing more concise (not just shorter, but rather to find effective phrasings so to say more in fewer words, so to then make space to additional written content), would be really helpful.

**Related works and baselines**: even if you claim attention to be inferior than a new approach, you still need to show experiments that empirically validate that's the case. All claims should always be empirically or concretely supported. Same for the related work section, even if prior works don't achieve your desiderata, you still need to discuss them, compare to them, explain how you are similar and different, what you do they don't, what prior works inspired or were the basis for your new idea, etc.

I therefore would unfortunately keep my score.

p.s. Thank you for the comment regarding "investigate in how far", I wasn't aware and am happy to learn about it!

**Strengths:**

- **Idea**: The paper proposes new neural structures and does a good job explicitly listing each of the specific design goals they seek to realize through their approach.
- **Topic**: Its research objectives of OOD generalization and ability to perform algorithmic tasks are important, and I personally think, under-explored themes in current deep learning research.
- **Evaluation**: The paper studies that idea through a nice variety of experiments of different kinds: addition and multiplication, double-addition, and algorithmic tasks.
- **Discussion**: There are good discussion and limitations sections that cover different aspects of the presented approach (computational efficiency, training stability, interpretability and scalability).

**Weaknesses:**

- **Semantic Grouping**: The paper claims that  block operations group the neurons into semantic units, but the grouping is actually done arbitrarily. The paper doesn’t provide evidence for semantic grouping or disentanglement to properties etc, quantitatively or qualitatively.
- **Clarity**: The writing, clarity and overall presentation could be improved significantly. This is especially the case up to page 5. Instead of having coherent sections with conjunctions and transitions, the paper is organized as lists of bullets or disconnected paragraphs, and in many cases not enough details are provided on each of them. Personally I think this form of succinct writing is less suitable for a paper, and would encourage the authors to work on the writing further both in order to increase its flow and coherency, and also to extend it to include more details. In the questions/suggestions section I refer to specific content that is missing through the different sections.
- **Terminology**: Multiple terms are used without being defined clearly and precisely enough: e.g. routing, uniform representations, representation-preserving mappings, copy gate. A notable additional example is that of negative interference, a term that is mentioned repeatedly through the paper but isn’t explained or illustrated. A further discussion, formal definitions or concrete examples could help clarify the mentioned terms.
- **Datasets**: The experiments in the paper are performed over synthetic discrete data only. I would suggest either exploring multiple modalities or real-world data to strengthen the empirical evaluation.
- **Baselines**: The experiments compare the new approach only to FNNs as baselines. Comparing to other approaches, such as recurrent networks and transformers, is in my opinion critical.
- **Novelty**: The authors contrast between self-attention and the new multiplexers idea, mentioning the strength of multipliers allowing for different “inputs and outputs”. These could be modeled by cross attention (which is broadly used too but isn’t mentioned in the paper). I think both a discussion of the conceptual differences as well as empirical head-to-head comparison between the proposed idea and cross attention will be very important.
- **Background & Related Works**: Not enough context is provided on the current issues of FNNs, the prior attempts to address them, what are their specific limitations that the paper seeks to address. For instance, where the paper mentions that a prior work “calls for additional research on suitable inductive biases” – what prior inductive biases have been explored in the literature, how is your approach different from them, etc. Likewise, the related works section mentions Capsule Networks, Routing Networks and Recurrent Independent Mechanisms, but provides a particularly brief description of them, and doesn’t in which ways they are similar or different than the new approach. I would also recommend discussing slot attention and its extensions.
- **Implicit Grouping**: It could be that FNNs perform some degree of grouping implicitly, even without externally imposing the grouping through inductive biases. In hierarchical deep networks it’s likely to be the case. It would be therefore good to either show empirically that FNNs do not do that effectively enough on the tasks that you study, or rephrase/tone down the paper’s claims about FNNS.

**Questions:**

- The paper says: “fully-connected layers treat all neurons as independent of each other”. I disagree with that on several levels. Since they have non-linearities, FNNs can model arbitrarily complex functions that do account for various subtle relations between the activations. They do the opposite of treating neurons as independent factors, and in fact research efforts on e.g. disentanglement aim to encourage them to treat latent factors more independently then what they do by default. In another level, models that are based on slots, tokens, key-value, or in the case of transformers heads, already incorporate inductive biases for grouping of activations into larger units. It would be good to discuss these in the paper.
- Section 3 says: “there is no inductive bias that would cause FNNs to keep activation patterns for the same concept consistent throughout the network.”. That’s a very strong statement that I’m not sure if there’s a way to prove. I would recommend rephrasing it.
- The last sentence in the abstract isn’t clear: “Adapting more complex architectures than the FNN to make use of them could lead to…” <-- are them the FNNs that are altered, or the current proposed idea, or future structural extensions?
- Second introduction’s paragraph:  investigated in how -> investigated how
- In section 3: “more dense” -> “denser”.

---

> ### Author Response · Authors · 2023-11-16
>
> We will address your complaints in order, before answering the questions:
>
> **Semantic Grouping**: We should clarify what we meant. As you say, The neurons are not grouped semantically in advance. However, the network has an inductive bias that causes the groups to take on semantically distinct roles. The experiments confirm this assumption.
>
> **Clarity**: Not sure what to say here. Different people have different preferences? We have had positive feedback about the bullet-point style before.
>
> **Terminology**: We would have liked to give more details on the things you mentioned, but we are constrained by the page limit. Explaining new ideas took priority over explaining things.
>
> **Datasets & Baselines**: The focus of this paper is on the core idea of Block Operations. Since the SMFR replaces the FNN, we only compare with it and run only tests that evaluate what we want to test. Real life tests are left as future work. As pointed out in the paper, even if SMFRs in their current state of development should perform poorly, our results show that the underlying idea of block-operations works as intended and is worth exploring further.
>
> **Novelty**: Attention models of all types are mathematically similar to SMFRs, but they have one crucial difference that makes a comparison pointless: Attention models are still connected to each other with fully-connected layers. These re-introduce the difficulties in data transfer that the SMFR sets out to solve.
>
> **Background & Related Work**: The reason we do not talk about these in detail is because none of them address the design goals mentioned in Section 3.1. We just mention that they exist to show that we are aware of them, because several of them look similar at first glance but they all fail to achieve the goals outlined in 3.1, usually because they were designed for different purposes.
>
> **Implicit Grouping**: As shown by Csordás et al. (2020), even if implicit groupings arise, they are not maintained throughout multiple layers of the network.
>
> **Questions**
>
> **On “fully-connected layers treat all neurons as independent of each other”**: We will phrase this differently. What we meant was that neurons are treated as independent by the learning process. I.e. their index within the tensor has no effect. Once learning has taken place, the meanings they have taken on are not independent, as you say. The existing models you mention that do group neurons into larger layers perform these groupings for different reasons that do not address our design goals in section 3.1.
>
> **On “there is no inductive bias that would cause FNNs to keep activation patterns for the same concept consistent throughout the network.”**: This is a finding of Csordás et al. (2020).
>
> **About the abstract**: "Them" refers to block-operations.
>
> **Typos**: The phrase "investigate in how far" is valid English and means "investigate to what extent". "More dense" should indeed be replaced with "denser". Thank you for pointing this out!

---

> ### Comment · Reviewer_VWus · 2023-12-04
> **Thanks for the comments**
>
> Dear authors, thanks for the comments. Some responses to them:
>
> **Writing/organization**: of course there could be preferences for different people on that, but personally I think it goes beyond writing style and damages the paper's clarity. It makes explanations kind of shorter, less detailed, and more disconnected. E.g. in slides when things are organized in a way more similar to that, the speaker does the job of making the presentation coherent, and so in the context of a paper, I personally believe this is less suitable / damages it. Indeed, you agree that you would like to add more details but couldn't due to page limit. I think more efforts on making the writing more concise (not just shorter, but rather to find effective phrasings so to say more in fewer words, so to then make space to additional written content), would be really helpful.
>
> **Related works and baselines**: even if you claim attention to be inferior than a new approach, you still need to show experiments that empirically validate that's the case. All claims should always be empirically or concretely supported. Same for the related work section, even if prior works don't achieve your desiderata, you still need to discuss them, compare to them, explain how you are similar and different, what you do they don't, what prior works inspired or were the basis for your new idea, etc.
>
> I therefore would unfortunately keep my score.
>
> p.s. Thank you for the comment regarding "investigate in how far", I wasn't aware and am happy to learn about it!

---

### Meta-Review · Area_Chair_zcn5 · 2023-12-04

**Metareview:**

This paper proposes two new architectural components: the Multiplexer and the FNNR. These act on groups of activations inside a layer (i.e. blocks of neurons) with the goal of providing a way for internal information routing that aids computation reuse and modularity. The benefit of these components are demonstrated empirically on addition, multiplication and algorithmic tasks. An improvement in terms of OOD performance compared to standard FNNs are shown.

The expert reviewers highlighted several strengths of this work, such as the relevance of the problem that is being studied, and the overall design of the new components. A clear discussion of limitations is also something that stood out. At the same time a number of issues were pointed out by the reviewers, such as the missing comparison to alternative baselines (eg. Transformers and recurrent models), overclaiming the novelty of some aspects of the approach, and the way the analysis was carried out, i.e. not all claims are properly supported. Finally, there are some concerns about the clarity of the paper, though this is minor.

Based on this, I tend to agree with the reviewers that this work is not currently ready for publication. A number of concrete suggestions for improving the paper were made, in particular concerning the baseline comparison and the datasets on which are being compared, and I would encourage the authors to implement these. While the current submission contributes some useful insights, without said comparisons and an overall improvement in the analysis, the significance is too limited.

**Justification For Why Not Higher Score:**

Too limited significance due to comparison only to FFNNs and slightly too simplistic datasets. Other concerns about the way the analysis was carried out are valid too, though they can be addressed more easily.

**Justification For Why Not Lower Score:**

N/A

---

### Decision · Program_Chairs · 2024-01-16

Reject